# Comparison of Mutated KRAS and Methylated HOXA9 Tumor-Specific DNA in Advanced Lung Adenocarcinoma

**DOI:** 10.3390/cancers12123728

**Published:** 2020-12-11

**Authors:** Sara W. C. Wen, Rikke F. Andersen, Lena Marie S. Petersen, Henrik Hager, Ole Hilberg, Anders Jakobsen, Torben F. Hansen

**Affiliations:** 1Department of Oncology, Vejle Hospital, University Hospital of Southern Denmark, Beriderbakken 4, 7100 Vejle, Denmark; anders.jakobsen@rsyd.dk (A.J.); torben.hansen@rsyd.dk (T.F.H.); 2Institute of Regional Health Research, University of Southern Denmark, 5230 Odense, Denmark; 3Department of Clinical Biochemistry, Vejle Hospital, University Hospital of Southern Denmark, Beriderbakken 4, 7100 Vejle, Denmark; rikke.fredslund.andersen@rsyd.dk; 4Department of Pathology, Vejle Hospital, University Hospital of Southern Denmark, Beriderbakken 4, 7100 Vejle, Denmark; lena.marie.skindhoj.petersen@rsyd.dk (L.M.S.P.); henrik.hager@rsyd.dk (H.H.); 5Department of Medicine, Vejle Hospital, University Hospital of Southern Denmark, Beriderbakken 4, 7100 Vejle, Denmark; ole.hilberg@rsyd.dk

**Keywords:** Gene methylations, gene mutations, circulating tumor DNA, HOXA9, KRAS, non-small cell lung cancer, lung adenocarcinoma

## Abstract

**Simple Summary:**

Lung cancer causes the largest number of cancer-related deaths worldwide. Circulating tumor DNA (ctDNA) has been suggested as a diagnostic and prognostic biomarker in non-small cell lung cancer, but the optimal target for measuring ctDNA has not been established. We aimed to compare a gene methylation biomarker with a gene mutation biomarker in order to determine the mutual agreement. Mutation analysis requires a broad and expensive test like next-generation sequencing, while methylation analysis can be performed by the less expensive droplet digital PCR. We found a good correlation between methylated HOXA9 and mutated KRAS in plasma from patients with lung adenocarcinoma.

**Abstract:**

Circulating tumor DNA (ctDNA) has been suggested as a biomarker in non-small cell lung cancer. The optimal target for measuring ctDNA has not yet been established. This study aimed to investigate methylated Homeobox A9 (meth-HOXA9) as an approach to detect ctDNA in advanced lung adenocarcinoma and compare it with mutated Kirsten rat sarcoma viral oncogene homolog (mut-KRAS) in order to determine the mutual agreement. DNA was purified from formalin-fixed, paraffin-embedded non-malignant lung tissue and lung adenocarcinoma tissue, and plasma from healthy donors and lung adenocarcinoma patients, respectively. KRAS mutations in tumor tissue were identified by next-generation sequencing and quantified in tumor and plasma by droplet digital polymerase chain reaction (ddPCR). The meth-HOXA9 analysis was based on bisulfite-converted DNA from tumor and plasma and quantified by ddPCR. Samples consisted of 20 archival non-malignant lung tissues, 48 advanced lung adenocarcinomas with matched plasma samples, and 100 plasma samples from healthy donors. A KRAS mutation was found in the tumor in 34/48 (70.8%) adenocarcinoma patients. All tumors were positive for meth-HOXA9, while none of the non-malignant lung tissues were. Meth-HOXA9 was detected in 36/48 (75%) of plasma samples, and the median level was 0.7% (range of 0–46.6%, *n* = 48). Mut-KRAS was detected in 29/34 (85.3%) of the plasma samples, and the median level was 1.2% (range of 0–46.1%, *n* = 34). There was a good correlation between meth-HOXA9 and mut-KRAS in plasma (Spearman’s rho 0.83, *p* < 0.001). Meth-HOXA9 is present in tissue from incurable lung adenocarcinoma but not in non-malignant lung tissue. It may be used as an approach for detecting ctDNA. The results demonstrated a high agreement between meth-HOXA9 and mut-KRAS in patients with advanced lung adenocarcinoma.

## 1. Introduction

Lung cancer causes the largest number of cancer-related deaths worldwide [1], and the prognosis remains poor with a 5-year survival rate of less than 20% [2,3]. Survival may be improved by implementing biomarkers in areas such as screening and treatment monitoring.

Circulating tumor-specific DNA (ctDNA) has been suggested as a biomarker in a wide range of malignant tumors. The obvious advantages are the easy access, the overcoming of tumor heterogeneity, the minimal discomfort to the patient, and the possibility of serial measurements. It is most commonly measured in a blood sample, but other body fluids such as urine, sputum, or bronchial lavage fluid have also been suggested [4,5,6]. ctDNA can be detected as a tumor-specific somatic mutation, for example, an oncogenic driver mutation or a tumor-specific aberrantly methylated gene. There is no consensus on the best approach to measuring ctDNA, and the method is not yet widely used in routine clinical practice [7,8].

Mutations can vary between individuals, but also between the original tumor and its metastases, or in the tumor over time [9,10]. Clonal hematopoiesis may also represent a problem causing false-positive findings. Analyzing mutations requires a broader screening test, like next-generation sequencing (NGS), to identify the relevant mutation to follow over time, the so-called “tumor-informed” approach. Alternatively, a broad test must be applied every time.

One of the most commonly mutated genes in lung cancer is the Kirsten rat sarcoma viral oncogene homolog (KRAS), which is mutated in around 30% of all lung adenocarcinomas [11,12]. Oncogenic mutation of KRAS causes its product, K-Ras, to be constitutively activated. Its downstream pathways regulate cell proliferation, differentiation, and survival [13]. Mutations most commonly occur in codon 12 but can also occur in codon 13, and they are associated with resistance to epidermal growth factor receptor (EGFR)-targeted therapy [13].

Another approach to the measurement of ctDNA is aberrant gene methylations [14]. Epigenetic changes, such as methylation of the promoter region, affect the transcription of a gene by adding or removing methyl groups from the DNA molecule. Hypermethylation usually inhibits gene transcription, while hypomethylation increases transcription [15]. Aberrant methylation of the promoter region of a tumor suppressor gene can contribute to cancer development [16]. Some promoter methylations have been shown to be persistent throughout disease stages [17,18], making them a relevant target for ctDNA detection.

The Homeobox family contains 39 genes located on four different chromosomes, and they share a common DNA sequence [19]. The Homeobox A9 (HOXA9) gene encodes a DNA-binding transcription factor, but its function is not fully elucidated. HOXA9 was significantly downregulated in breast cancer tissue compared with normal breast tissue [20], and the gene’s promoter region was found to be hypermethylated in ovarian cancer [21]. Downregulation of HOXA9 has been suggested to enhance the migratory potential of lung cancer cells [22] and to stimulate cell invasiveness as a target of miR-196b [23]. Hypermethylation of HOXA9 (meth-HOXA9) has been shown to be both a sensitive and specific diagnostic marker in lung cancer tissue [24,25]. It has also been suggested for both early detection and as a prognostic marker in blood samples [4,26].

The main challenge when detecting mutated ctDNA is that there are many different driver mutations, but none are universally found in the majority of lung tumors. On the contrary, some aberrantly methylated genes have been found in 59–90% of tumor samples [25,26]. A biomarker found in a large fraction of tumors may have wider clinical utility. Therefore, it would be relevant to compare one of the frequently mutated genes, KRAS, with one of the frequently methylated genes, HOXA9.

The aim of the present study was to investigate methylated HOXA9 as an approach to detect ctDNA in advanced lung adenocarcinoma. Specifically, we wanted to (i) assess the prevalence of methylated HOXA9 in malignant and non-malignant lung tissue and plasma samples, (ii) establish a cutoff in tissue and plasma to discriminate between lung adenocarcinoma and non-cancer subjects, and (iii) investigate the correlation between methylated HOXA9 and mutated KRAS in tissue and plasma.

## 2. Results

### 2.1. Cohort Characteristics

The median age for the healthy donors was 53 years (range of 19–82, *n* = 81), while the patients with incurable lung adenocarcinoma had a median age of 71 years (range of 53–82, *n* = 48, *p* < 0.001). Among the cancer patients, three had stage III disease, while the rest had stage IV disease. There were significantly more females among the patients with lung adenocarcinoma (72.9%, *n* = 48) compared with the healthy donors (56.0%, *n* = 100, *p* = 0.048).

### 2.2. HOXA9 Methylation in Lung Tissue

Non-malignant lung tissue had a median of 0.4% meth-HOXA9 (range of 0.2–0.7%, *n* = 20), while lung adenocarcinoma tissue had a median of 13.9% meth-HOXA9 (range of 0.8–75%, *n* = 48, *p* < 0.001) (see Figure 1).

Receiver operating characteristic (ROC) curve analysis (Figure 2) of malignant and non-malignant tissues resulted in an area under the curve (AUC) of 1.0 (95% CI 1.0-1.0, *n* = 68). The optimal cut-point for meth-HOXA9 was established at ≥0.8%, resulting in perfect discrimination between malignant and non-malignant lung tissue.

### 2.3. HOXA9 Methylation in Plasma

The digital droplet polymerase chain reaction (ddPCR) analysis detected meth-HOXA9 in 2/100 (2%) of plasma samples from healthy donors, as described in Materials and Methods paragraph 4.5, and 36/48 (75%) of plasma samples from patients with incurable lung adenocarcinoma. This corresponds to a sensitivity of 75% and a specificity of 98%. The median meth-HOXA9 level was 0% (range of 0–0.53%, *n* = 100) in donors and 0.7% (range of 0–46.6%, *n* = 48, *p* < 0.001) in lung adenocarcinoma patients (Figure 1).

### 2.4. KRAS Mutations in Lung Adenocarcinoma Tissue

KRAS mutations were detected by NGS in tumor tissue from 34/48 (70.8%) of patients with lung adenocarcinoma. The mutations were G12C (15/34), G12D (6/34), G12V (6/34), G12A (3/34), G12R (2/34), G12S (1/34), and Q61H (1/34). The identified mutations were then analyzed by ddPCR in purified DNA from the same tumors. The median level of mut-KRAS was 14.7% (range of 2.1–62.3%, *n* = 34) (Figure 1). KRAS mutations were not analyzed in tissue from the patients providing non-malignant lung tissue.

### 2.5. KRAS Mutations in Plasma

The KRAS mutations identified by NGS in tumor tissue were analyzed by ddPCR in matched plasma samples. Mut-KRAS was detected in 29/34 (85.3%) of the plasma samples, and the median level was 1.2% (range of 0–46.1%, *n* = 34) (Figure 1).

### 2.6. Correlation Between Methylated and Mutated ctDNA

Meth-HOXA9 and mut-KRAS were measured by ddPCR in DNA from tumor and plasma from 34 patients with incurable lung adenocarcinoma. The analyses from tumor DNA detected both meth-HOXA9 and mut-KRAS in each tumor, but the levels varied. The Spearman rank correlation was moderate (Spearman’s rho = 0.61, *p* < 0.001, Figure 3A).

In plasma, the results were discordant in five cases: four patients had detectable mut-KRAS but not meth-HOXA9, while one patient had the opposite. The Spearman rank correlation is shown in Figure 3B (Spearman’s rho = 0.83, *p* < 0.001).

## 3. Discussion

In this retrospective analysis of lung tissue and plasma, we found that meth-HOXA9 had high sensitivity and specificity for diagnosing advanced lung adenocarcinoma. Plasma analysis indicated a high correlation between meth-HOXA9 and mut-KRAS.

We found 100% of the lung tumors to be positive for meth-HOXA9 using the cut-point of 0.8%, as described. This is in range with findings by Wrangle et al. [25], who reported meth-HOXA9 with a sensitivity of 97% in lung cancer data from the Cancer Genome Atlas, and 90% and 73%, respectively, in independent cohorts from the United States and Japan. Nawaz et al. [24] reported a sensitivity of 87% for meth-HOXA9 in tumor tissue. Ooki et al. [26], however, reported a sensitivity of only 59% in 90 tumor samples from American patients with non-small cell lung cancer (NSCLC). Our cohort was exclusively Caucasian and consisted of advanced lung adenocarcinoma, whereas both the American and Japanese cohorts included different races, stages, and histologic types of NSCLC. The Cancer Genome Atlas used methylation array, while Wrangle et al., Nawaz et al., and Ooki et al. used quantitative methylation-specific PCR (Q-MSP), and we used ddPCR. These different methods could also explain some of the observed differences, as they differ with regard to sensitivity.

Generally, around 30% of lung tumors—primarily adenocarcinomas—harbor a KRAS mutation [11,12]. We found a frequency of 70.8% in our cohort of patients with incurable adenocarcinoma treated with immunotherapy. The higher mut-KRAS frequency could be due to the fact that our cohort consisted solely of adenocarcinomas from Caucasian individuals treated with checkpoint inhibitor immunotherapy, thus excluding the patients with targetable mutations [27]. The mut-KRAS frequency was relatively high in our cohort, but the frequency of meth-HOXA9 was even higher. This suggests that meth-HOXA9 may be found more commonly in advanced lung adenocarcinomas than mut-KRAS. Our results are also in line with the findings of Nawaz et al. [24] and Wrangle et al. [25], as outlined above. Since the larger fraction of lung cancer patients is KRAS wild type, it is likely that methylated HOXA9 can be found in lung cancer patients irrespective of KRAS status.

We detected meth-HOXA9 with a sensitivity of 75.0% and a specificity of 98.0% in plasma from patients with advanced lung adenocarcinoma. Liu et al. [6] reported a sensitivity and specificity for plasma meth-HOXA9 of 58% and 80%, respectively, in patients suspected of NSCLC. Ooki et al. [26] reported quite different results with a sensitivity and specificity of 27.9% and 92.9%, respectively, in serum from patients with stage I NSCLC. These big differences represent a challenge in the development of methylated ctDNA (meth-ctDNA) as a biomarker. DNA extraction and analysis method are important factors [28]. Both Liu et al. and Ooki et al. used Q-MSP, while we used methylation-specific ddPCR. We used plasma, while Ooki et al. used serum. This may explain some of the observed differences, as ddPCR is a more recent technique and is considered highly sensitive [29]. The patient cohorts were different regarding race and histology, which may cause different results. Ooki et al. only included stage I patients in the serum analysis; this may be the reason for the much lower sensitivity reported. One approach to increasing diagnostic accuracy is to employ a panel of biomarkers and use criteria such as “any of three markers positive” or calculate a risk score based on the number of positive markers. This is a very common approach and often increases sensitivity [6], but it can potentially decrease specificity [26].

We found a good correlation between mut-KRAS and meth-HOXA9 in plasma. The analyses were concordant in all but five cases. This could be because the marker was truly non-existent in some patients; however, meth-HOXA9 was detected in all tumor samples which makes this unlikely. It could be due to the tumor releasing no or very small amounts of DNA into the circulation. This has been shown to depend on clinicopathological features, such as tumor size, necrosis, or mitosis frequency [30]. It is also likely that the methylation analysis was less sensitive because of the bisulfite conversion step. Conversion causes loss of DNA, indicating that larger amounts of plasma may be needed for analysis of meth-ctDNA compared with mutated ctDNA (mut-ctDNA).

Meth-ctDNA has already been suggested as a universal cancer-specific biomarker in previous studies [14,17,31]. The present study finds that meth-HOXA9 is a sensitive and specific biomarker of advanced lung adenocarcinoma in tumor tissue, and it has promising qualities for use in liquid biopsies. Analysis of mut-KRAS needs to be tumor-informed or performed with NGS, whereas meth-HOXA9 can be analyzed directly in a simple, fast, and cheap ddPCR assay. Meth-ctDNA does not, however, give any information about targetable mutations, which is highly important at the time of diagnosis or when the patient develops drug resistance. Nevertheless, meth-HOXA9 may have potential as a diagnostic biomarker since it seems to be more prevalent in lung tumors than mut-KRAS. It may also hold prognostic value as previously suggested [26]. Given its high prevalence, meth-HOXA9 may also hold potential in treatment monitoring, but this remains to be investigated in future studies.

The main weakness of the present study was the modest patient cohort of 48 patients, with only 34 available for the comparison of mut-KRAS and meth-HOXA9. Likewise, the number of lung tissue controls (*n* = 20) was small. Given that there was no overlap in the levels of meth-HOXA9 between the non-malignant tissue samples and the tumor tissue samples, however, we believe that the sample size for this purpose is reasonable. We did not analyze mut-KRAS on the non-malignant lung tissue samples and plasma samples from the healthy donors enrolled in this study. Such analyses could potentially serve as an adjunct to those presented here. Mut-KRAS in tumor tissue was initially analyzed by NGS and then by ddPCR. Meth-HOXA9 was only analyzed by ddPCR. Optimally, the two markers should be analyzed by both methods, but the many different KRAS mutations complicate the use of ddPCR, as does the initial method. As discussed above, the DNA input for the meth-HOXA9 analysis could have been larger in order to increase sensitivity.

## 4. Materials and Methods

### 4.1. Participants

Non-malignant lung tissues were archival specimens obtained from patients having lung biopsy performed during diagnostic workup for diseases other than lung cancer. Lung cancer tumor tissue was obtained from patients with histopathologically confirmed adenocarcinoma originating from the lung. Further inclusion criteria were eligibility for treatment with checkpoint inhibitor immunotherapy in any treatment line, age ≥ 18 years, and the ability to understand and follow protocol guidelines. Exclusion criteria comprised other experimental treatment within 14 days from treatment initiation and pregnancy or breastfeeding.

Non-malignant plasma was collected from healthy individuals.

The study was carried out according to the rules of the Declaration of Helsinki of 1975, revised in 2013, and the Danish data protection legislation. All participants provided written, informed consent before enrolling in the study. The study protocol was approved by the Regional Committee for Health Research Ethics of Southern Denmark (ID number S-20170063, 8 June 2017).

### 4.2. Samples

The study included 68 lung tissue samples divided into subjects without lung cancer (*n* = 20) and patients with incurable lung adenocarcinoma receiving checkpoint inhibitor immunotherapy (*n* = 48). These patients also provided matching plasma samples (*n* = 48), which were drawn before treatment with immunotherapy was initiated. We obtained plasma samples from healthy donors (*n* = 100) as controls. An overview is provided in Figure 4.

### 4.3. DNA Extraction

Cell-free DNA was extracted from 2 × 4 mL of plasma, as previously reported [32], using the DSP Circulating DNA kit (Qiagen, Hilden, Germany) according to the manufacturer’s instructions. The β2 microglobulin gene was used as a surrogate for the total amount of cell-free DNA.

Genomic DNA was extracted from formalin-fixed, paraffin-embedded (FFPE) lung tissue (tumor or non-tumor), as previously described [31]. In tumor tissue, areas with a high degree of tumor cells were selected by an experienced pathologist. Depending on tissue content, 1–10 slices with a thickness of 10 μm were immersed in 180 μL incubation buffer and 20 μL protein kinase K at 70 °C overnight. DNA was released by adding 400 μL lysis buffer and purified with the Maxwell 16 FFPE Plus LEV DNA Purification kit (Promega, Madison, WI, USA). The samples were purified on a MAXWELL™16 LEV instrument (Promega, Madison, WI, USA). DNA was eluted in 500 μL nuclease-free water.

### 4.4. HOXA9 Methylation Analysis

The HOXA9 methylation-specific analysis was previously reported by our group [33]. Briefly, extraction of plasma DNA was assessed by spike-in of an internal control (cysteine-rich polycomb-like protein 1, CPP1, a gene from soybean, [34]). DNA was bisulfite-converted using the EZ DNA Methylation-Lightning Kit (Zymo Research, Irvine, CA, USA), as recommended by the manufacturer. The converted DNA was analyzed with an in-house HOXA9 methylation-specific ddPCR assay [32,33] with albumin as the reference gene. Water served as the negative control, a pool of lymphocyte DNA from healthy donors as the non-cancer control, and Universal Methylated DNA Standard (Zymo Research, Irvine, CA, USA) as the positive control. These controls were bisulfite-converted and analyzed by ddPCR, alongside the patient samples. Results were read on a QX200 Droplet Digital Reader (Bio-Rad, Hercules, CA, USA).

### 4.5. Methylated HOXA9 Limit of Blank and Limit of Detection

Plasma from the healthy donors was used for establishing the limit of blank. They were split into a discovery cohort and a validation cohort. Plasma from the 50 donors in the discovery cohort was analyzed for meth-HOXA9. Results were exported from the QX200 Droplet Digital Reader (Bio-Rad, Hercules, CA, USA) as the number of droplets containing meth-HOXA9 detected by ddPCR, and a <5% false-positive rate was arbitrarily decided as realistic. This resulted in a cut-point of ≥5 droplets containing meth-HOXA9 equaling a positive test, as this was true of one healthy donor (2%). The validation cohort of the 50 other donors showed exactly the same results of one donor with ≥5 droplets containing meth-HOXA9. Hence, the limit of blank was set at 4 droplets, and the limit of detection was set at 5 droplets.

After setting the limit of blank, data were normalized to the level of the albumin gene. Meth-HOXA9 copies divided by albumin copies resulted in a fraction of meth-HOXA9. These data were exported from QuantaSoft™ (Bio-Rad, Hercules, CA, USA) as the percentage of meth-HOXA9, including a 95% confidence interval (CI) derived from a Poisson distribution [35].

### 4.6. KRAS Mutations Analyzed by NGS

Genomic DNA was extracted from FFPE tumor tissue, as described above. KRAS mutations in tumor tissue were analyzed by NGS with the TruSight^®^ Tumor 15 kit (Illumina, San Diego, CA, USA). The DNA libraries for sequencing were prepared according to the instructions in the TruSight^®^ Tumor 15 Reference Guide [36]. Briefly, the samples were identified with a unique tag before pooling, and then the genes of interest were captured and amplified on a Veriti™ Thermal cycler (Applied Biosystems, Foster City, CA, USA). Clusters were generated on a flow cell by bridge amplification, including a control library (PhiX, Illumina, San Diego, CA, USA), and then sequenced by paired-end sequencing on the MiniSeq^®^ System (Illumina, San Diego, CA, USA). Data were analyzed using the on-instrument MiniSeq System Software. Variant calling required ≥500x coverage for detecting mutations with a frequency of ≥2%.

### 4.7. KRAS Mutations Analyzed by ddPCR

KRAS mutations were analyzed on DNA from the tumor samples as well as on plasma DNA. The ddPCR method was previously reported [31]. The analysis was performed as described above for meth-HOXA9, except for the bisulfite conversion of the DNA. We used PrimePCR ddPCR assays (Bio-Rad, Hercules, CA, USA) for the specific mutations in the tumor tissue harbored by the patient, as detected by NGS.

### 4.8. KRAS Mutation Limit of Blank and Limit of Detection

The limit of blank in plasma was only established for the KRAS G12D mutation, which is one of the common mutations. Plasma from healthy donors (*n* = 20) was analyzed by ddPCR, and <2 positive droplets were observed in all analyses [31]. A pool of lymphocyte DNA from healthy donors was analyzed for all KRAS mutations, and <3 mutation-positive droplets were found in all analyses. Hence, the limit of blank was set at 2 droplets, and the limit of detection was set at 3 droplets.

### 4.9. Statistical Analyses

Data are presented as frequency (percentage) for categorical variables and as median (range) for continuous variables. Comparisons between two groups were made with the Wilcoxon rank-sum test for numeric variables and Pearson’s Chi-squared test for categorical variables. Diagnostic accuracy was analyzed by ROC analysis. Correlation between meth-HOXA9 and mut-KRAS was examined with the Spearman Rank Correlation. All tests were two-sided; *p*-values < 0.05 were considered significant. All analyses were performed using STATA 16IC (StataCorp LLC, TX, USA).

## 5. Conclusions

In conclusion, meth-HOXA9 is present in tissue from incurable lung adenocarcinoma but not in non-malignant lung tissue. It may be used as an approach for detecting ctDNA. The results demonstrated a high agreement between meth-HOXA9 and mut-KRAS in patients with advanced lung adenocarcinoma. Future research should focus on meth-HOXA9 in relation to diagnosis, prognosis, and treatment monitoring.

## Figures and Tables

**Figure 1 cancers-12-03728-f001:**
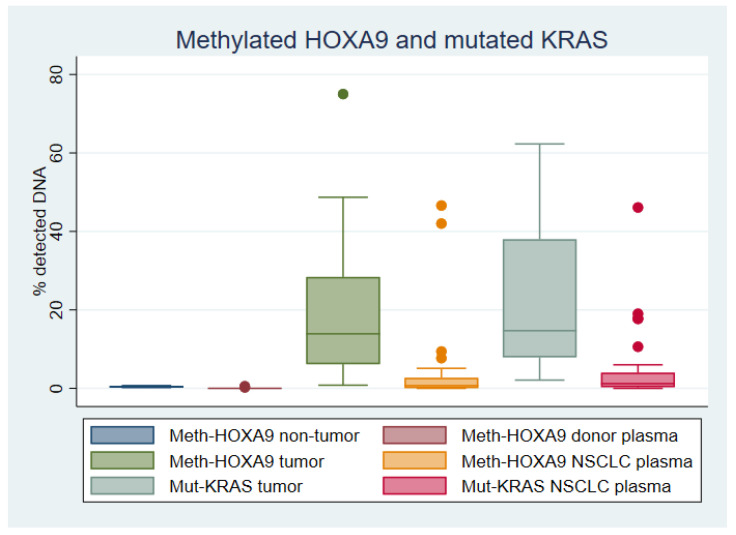
Levels of methylated Homeobox A9 (meth-HOXA9) and mutated Kirsten rat sarcoma viral oncogene homolog (mut-KRAS) in tissue and plasma. Boxplot depicting the level of meth-HOXA9 in non-malignant lung tissue (*n* = 20), plasma from healthy donors (*n* = 100), lung adenocarcinoma tissue (*n* = 48), matched plasma from lung adenocarcinoma patients (*n* = 48), and level of mut-KRAS in lung adenocarcinoma tissue (*n* = 34) and matched plasma (*n* = 34).

**Figure 2 cancers-12-03728-f002:**
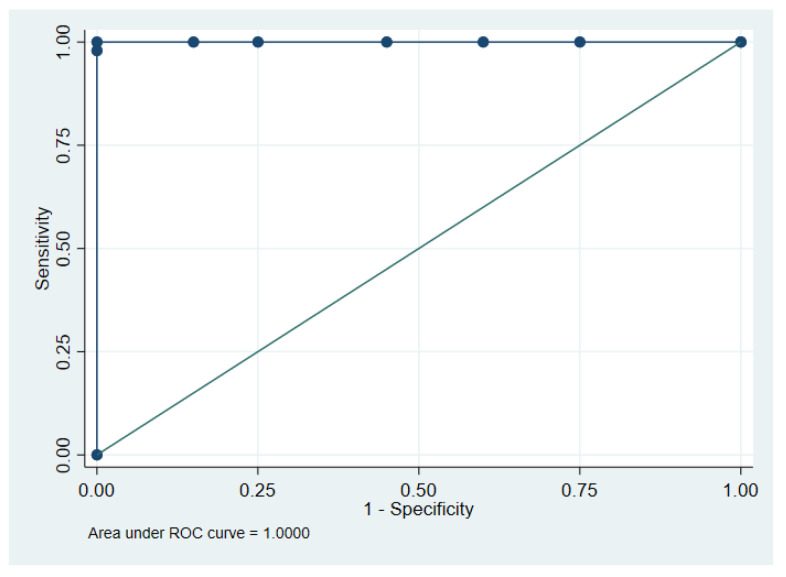
Methylated Homeobox A9 (meth-HOXA9) in lung tissue. Receiver operating characteristic (ROC) curve of meth-HOXA9 in malignant (*n* = 48) and non-malignant (*n* = 20) lung tissues.

**Figure 3 cancers-12-03728-f003:**
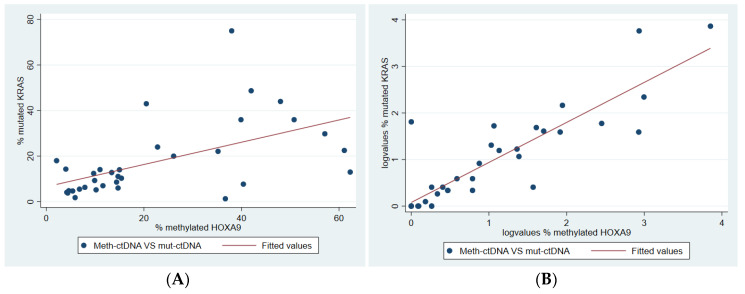
Correlation between methylated Homeobox A9 (meth-HOXA9) and mutated Kirsten rat sarcoma viral oncogene homolog (mut-KRAS). Two-way scatter plot of (**A**) tumor meth-HOXA9 percentages plotted against mut-KRAS percentages and a trend line and (**B**) plasma meth-HOXA9 percentages plotted against the mut-KRAS percentages (both log-transformed) with a trend line.

**Figure 4 cancers-12-03728-f004:**
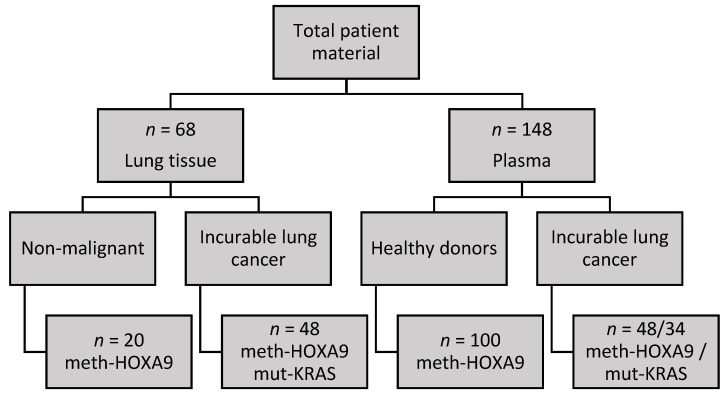
Biologic material. Flow chart illustrating the types of biologic samples available, number of samples analyzed, and which markers were analyzed in each cohort. Tumor and patient plasma samples were matched. Mutated Kirsten rat sarcoma viral oncogene homolog (mut-KRAS) was only analyzed in plasma from the patients with a KRAS mutation detected by next-generation sequencing (NGS) in tumor tissue.

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
