# Peer review of "Comparison of Mutated KRAS and Methylated HOXA9 Tumor-Specific DNA in Advanced Lung Adenocarcinoma"

_cancers, 2020, doi:10.3390/cancers12123728_

Round 1
Reviewer 1 Report
The authors have satisfactorily addressed my comments.
Reviewer 2 Report
The authors have addressed all comments/questions raised at the time of initial review.
This manuscript is a resubmission of an earlier submission. The following is a list of the peer review reports and author responses from that submission.
Round 1
Reviewer 1 Report
In this paper, the authors describe meth-HOXA9 detection in tissue and plasma samples obtained from patients with advanced lung adenocarcinoma treated with immune checkpoint inhibitor, and compare it with non-malignant lung tissue and plasma samples obtained from volunteers. The authors also correlate the detection of meth-HOXA9 with mut-KRAS in cancer tissue and plasma samples.
The manuscript is generally well-written and is related to the area of blood-based biomarkers in lung cancer, which has gained a significant traction in the recent years. However, there are some major limitations that the authors need to address.
- The paper does not provide enough background information on HOXA9- whether it is specific for NSCLC or detectable in other tumors, and why the authors decided to analyze its correlation specifically with KRAS mutation and not any other known and common mutation in lung cancer.
- The overall discussion and conclusion do not convey a clear message and a path forward towards application in clinical practice.
- While the cost difference between NGS of KRAS mutation vs ddPCR for meth-HOXA9 is a valid point, clinical relevance of this is somewhat questionable. In routine clinical practice ctDNA analysis is done to detect rather multiple mutations than just focusing on KRAS. Consequently, NGS – although expensive – is extremely helpful in treatment decision making. Meth-HOXA9 does not have any implications in treatment decisions, at least based on currently available evidence.
- The authors suggest that meth-HOXA9 could be used as a universal biomarker in advanced NSCLC regardless of KRAS mutation status. The presented data do not support the conclusion. Additionally the study included only lung adenocarcinoma samples, so the conclusion should not be generalized to non-small cell lung cancer.
- In line 215, the authors suggest that detection of meth-HOXA9 would be an easy assay to monitor patients during treatment or after surgery, but the data presented in the paper do not support this. There no information on the change in the level of meth- HOXA9 with treatment. Additionally, most of the discussion is focused on binary outcome of meth-HOXA9 (detectable vs not), but the information on the level and discussion regarding changes in the level is lacking. From the range of detection level, it appears that many patients have very low level of detectable meth-HOXA9 – which could present a significant issue when the method is used for monitoring purposes. There is no data presented on the correlation with disease burden.
- The paper does not provide information on the detection of meth-HOXA9 in KRAS wild type population. This is an important consideration since approximately 70% of patients with NSCLC are KRAS wt (per TCGA).
- The authors suggest in line 184-185 that patients treated with immune checkpoint inhibitors often have a smoking history, is not entirely accurate. While smoking has been linked with favorable response to ICI, smoking status/history does not play a role in clinical decision making.
Reviewer 2 Report
The authors studied Kras mutations and hypermethylated HOXA9 in tissue and plasma samples in 48 adenocarcinomas treated with immunotherapy, 20 benign lung tissue and in plasma from 100 healthy subjects. They found Kras mutations in over 70% of the tumors well above the average rate of 25-30%. This was due to enrichment caused by studying tumors treated with immunotherapy thus excluding tumors with targetable mutations. The authors found good correlation between Kras mutations and meth-HOXA9 in the same tumor samples. There were, however, 5 discrepant cases in plasma with 4 cases positive for kras mutation and negative for meth-HOXA9 and 1 case negative for Kras and positive for meth-HOXA9. The authors postulated that most likely this due to insufficiencies in the bisulfide conversion step. This poses a significant limitation on using the meth-HOXA9 as a surrogate marker for Kras mutation in plasma samples. Fuurthermore, the authors have concluded that meth-HOXA9 may be used as a universal biomarker in advanced NSCLC as it was detected in all tumor samples irrespective of K-ras status and was absent in benign lung tissues. While this is an interesting finding, the cohort was biased as it only included tumors that were treated with immunotherapy and excluded tumors with targetable alterations and squamous histology.
In the introduction the authors should better explain their rationale for choosing meth-HOXA9 as a surrogate marker.
They should also clarify if the 48 plasma samples are matched with the 48 tumor tissue samples or if they represent 48 different subjects. Testing non-neoplastic tissue from patients with tumor tissue samples for meth-HOXA9 would also be interesting.
The references are not numbered in the reference section of the manuscript.
Minor comments:
Introduction,
para 1. Liquid biopsy targeting ctDNA is approved for clinical use in the US in metastatic NSCLC and is being used in clinical practice.
Results,
Cohort characteristics: Please define incurable disease (what clinical stage was included).
Reviewer 3 Report
Dear Authors
Your study is interesting and even original. However , you will need to improve the introduction as well the bibliography.
Morevover you vill need to discuss more of your limitations. Now is determinat to find biomarkers for early stage and its translation in clinical practice so pleae keep in mind to discuss with this purpose.